# The Behavioral Adaptations and Barriers of Patients Employing Non-Pharmacological Strategies for Cancer Pain Management—A Qualitative Study

**DOI:** 10.3390/healthcare11222911

**Published:** 2023-11-07

**Authors:** Man-Ting Liu, Shu-Yuan Liang, Ta-Chung Chao, Ling-Ming Tseng, John Rosenberg

**Affiliations:** 1School of Nursing, National Taipei University of Nursing and Health Sciences, 365 Ming Te Road, Beitou, Taipei 112, Taiwan; 2Department of Nursing, Cardinal Tien Junior College of Healthcare and Management, 407, Section 2, Jianfu Road, Shangwu Village, Sanxing Township, Yilan 266, Taiwan; 3Oncology Department, Taipei Veterans General Hospital, 201, Sec. 2, Shipai Road, Beitou, Taipei 112, Taiwan; 4Department of Medicine, School of Medicine, National Yang-Ming University, Taipei 112, Taiwan; 5Department of General Surgery, Taipei Veterans General Hospital, 201, Sec. 2, Shipai Road, Beitou, Taipei 112, Taiwan; 6School of Health, University of the Sunshine Coast, Caboolture, QLD 4059, Australia; jrosenbe@usc.edu.au

**Keywords:** cancer, pain, non-pharmacological strategies, behavioral adaptations, qualitative study

## Abstract

The use of non-pharmacological strategies to complement pharmacological approaches can enhance cancer pain management by promoting patient autonomy and increasing management effectiveness. This study aimed to explore the required behavioral adaptations and situational barriers that cancer patients encounter when utilizing non-pharmacological strategies to manage pain. We adopted an exploratory–descriptive qualitative research approach, purposive sampling, and semi-structured interview guidelines to conduct face-to-face interviews with 18 cancer patients experiencing moderate or severe levels of worst pain. Data were analyzed using inductive content analysis to explore patients’ experiences. Five themes described the behavioral adaptations of patients using non-pharmacological strategies to deal with cancer pain: finding complementary therapies, utilizing assistive skills, adapting to assistive skills, diverting attention, and seeking help. Situational barriers faced by patients include being in the workplace or in a climate-affected environment. Behavioral adaptation is necessary for non-pharmacological strategies to coping with cancer pain. The behavioral skills can help the patients to overcome situational barriers to engagement with these strategies. Thus, health professionals are expected to help the patients acquire adequate behavioral adaptation and skills for self-pain management, and assess the effectiveness of the strategies.

## 1. Introduction

Cancer patients commonly face physical and mental suffering resulting from cancer diagnosis, treatment, and progression, and cancer pain is often the outcome that is most distressing and feared by patients [1]. Approximately 50% of cancer patients in Taiwan have experienced pain, and 45.7% of these patients have experienced poor pain relief [2]. Pain can disrupt patients’ sleep, cause fatigue, depression, anxiety, helplessness, and hopelessness, and incur higher medical care costs [3,4].

Even with advance in pharmacological interventions for cancer pain, cancer patients still report persistent pain [5,6]. While there are pharmacological approaches to cancer care, evidence has presented the limitations of pharmacological interventions, which includes high level of addiction to painkillers and side effects of pain medications [7]. These apprehensions may affect patients’ willingness to implement pharmacological strategies to manage their pain.

Pain is a complex domain of suffering including physical, psychological, social–cultural, and spiritual dimensions [8,9]. The WHO guidelines for managing cancer pain recommend a three-step analgesic ladder conceptual model to manage severe, moderate, and mild cancer pain. The three-step process combines pharmacological and non-pharmacological strategies [10]. A non-pharmacological strategy helps to reduce the required dosage of painkillers and to mitigate the side effects of medication [11] while increasing patients’ autonomy in pain management. The adult cancer pain guidelines developed by the National Comprehensive Cancer Network [12] also recommend the use of both pharmacological and non-pharmacological strategies, such as massage, acupuncture, music, and transcutaneous electrical stimulation [12], as well as psychosocial intervention such as meditation and faith-based treatments [10]. Studies have demonstrated that after controlling for medication, listening to music can significantly reduce postoperative pain in breast cancer patients [13], and massages also significantly reduce pain in patients [14]. Another randomized clinical trial (RCT) study involving patients with terminal cancer found that a combination of massages and exercise not only significantly reduced pain but also improved psychological discomfort [15]. Studies have also shown that acupuncture can improve many physiological symptoms experienced by cancer patients, and its most effective application is in pain relief [16]. The use of hot compresses was found to be effective for easing foot pain in patients [17]. Another study found that mindfulness meditation was significantly associated with a decrease in pain [18]. Liang et al. found that 56.9% of cancer patients chose to utilize both pharmacological and non-pharmacological strategies to manage their pain [19], highlighting the prevalence of non-pharmacological methods in cancer pain management.

Although non-pharmacological strategies can effectively provide relief for cancer-related pain, there is a possibility of complications during the implementation process. For example, massage therapy may lead to discomfort such as bruising, headache, and fatigue [20]. Patients may also experience difficulties in adopting non-pharmacological strategies such as physical activity, which could obstruct their pain management behaviors. For effective pain control, patients need to actively engage in pain management behaviors. As such, it is essential for medical professionals to understand any relevant challenges patients may face when using non-pharmacological approaches. The objective of the present study was to use semi-structured qualitative interview guidelines to gain insights into the major behavioral adaptations employed and situational obstacles faced by patients when implementing non-pharmacological strategies to manage cancer-related pain.

## 2. Methods

### 2.1. Research Design

This study adopted an exploratory–descriptive qualitative research approach, a purposive sampling method, and semi-structured interview guidelines to conduct in-depth interviews and collect patients’ narratives. Data were collected and analyzed to understand behavior adaptations and obstacles that patients encounter when managing their cancer pain through non-pharmacological strategies.

### 2.2. Sample and Setting

The study took place at an inpatient oncology ward in a medical center in northern Taiwan. It involved cancer patients who (1) had experienced pain for more than 3 months, (2) suffered from a worst pain with a rating of 4 points (inclusive) or more on a 0–10 scale in the past week, (3) currently and routinely used prescribed opioid analgesics, (4) were at least 20 years old, and (5) could communicate in Mandarin or Taiwanese. Further patient pain assessment used the Brief Pain Inventory [21]. In this study, a sufficient sample size of eighteen patients was determined based on the data saturation of findings; that is, data collection stopped when no new categories could be derived. This agrees with previous findings that qualitative studies most often require a sample size of 4–40 informants to satisfy the information-saturation criterion [22,23].

### 2.3. Data Collection Procedures and Data Analysis

The study protocol received approval from the hospital’s Institutional Review Board (IRB) (VGHIRB No. 2021-05-005A). Before data collection, the investigator first explained the study purpose, procedure, and the recording requirement of the interview process to the participants. The interviews were conducted after the participants provided informed consent and signed the consent form. Data collection occurred from 16 September 2021, to 30 November 2021. The present study used inductive content analysis to analyze the interview data with respect to categories and themes. We employed descriptive statistics including mean (M), standard deviation (SD), frequency (n), and percentage (%) to organize the patient’s demographic and medical characteristics.

The investigator collected data using semi-structured interview guidelines, such as “Could you please describe your pain management experience?”; “Could you please describe which non-pharmacological methods you have used to manage your pain?”; “How do you use non-pharmacological method A (B, C…) to manage your pain?”; ”How effective has it been to use non-pharmacological method A (B, C…)?”; “Could you please describe what problems or barriers you face when dealing with pain non-pharmacologically, and in what context in particular?”; ”What situations prevent you from using non-pharmacological methods?”; “How do you deal with barriers when using non-pharmacological methods to manage pain?”; “What is the effect of your response method A (B, C…) to deal with barriers for managing pain?”. One-on-one interviews were conducted in a private setting, with each interview lasting about 30–40 min. The investigator followed the semi-structured interview guidelines to guide participants in exploring their personal experience in depth. Throughout the interview process, the investigator maintained a neutral, empathetic, and open attitude and refrained from interjecting any subjective judgment.

After the interview, the recordings were transcribed verbatim and analyzed within 48 h. The data generated during the research process (audio recordings, transcriptions, and textual analysis) were retained to establish audit trails. As noted above, coding consistency between the two researchers was high, ensuring the reliability of this study. After the content analysis was completed, another researcher was invited to analyze and code 10–15% of the interview data and develop categories [24]. The agreement between the categories of the two researchers reached 80%, indicating that the themes or categories appropriately characterized the meaning of the data [25].

The rigor of this study was based on ensuring credibility, transferability, dependability, and confirmability [26]. Credibility refers to the extent to which research findings accurately reflect the subjective experience of the participants. In this study, the investigators underwent qualitative research-related training and established a trusting and close relationship with the participants, enabling the interviewees to openly express their experiences and provide original information for the study. The present study used a semi-structured interview guide to conduct interviews, and retained the interviewer’s field notes. The qualitative data analysis process was conducted by two researchers using mutual peer review. In addition, the results were subjected to a final review by the research team, which helped improve the validity of the research findings. To maintain an objective and neutral position and to overcome bias and avoid imposing personal opinions on participants, the investigators self-adjusted through reflection during the process to ensure suitable confirmability of the study. Furthermore, the audit trail was used to assess the accuracy of the findings and the truthfulness of the participants’ perspectives. Transferability refers to the extent that the research findings can be applied to similar contexts beyond the original research. This study clearly and in detail describes the study site where participants were recruited, inclusion criteria, and participant characteristics, as well as the data collection and analysis process, to facilitate future application in similar settings. For dependability, this study adopted purposive sampling to enroll participants with similar experiences. In addition, the investigators strengthened their interview skills before the study and paid attention to participants’ non-verbal behaviors during the interview process, such as facial expressions and gestures, which provided a reference for subsequent analysis. The description of the coding and the themes was also checked and reconfirmed by a researcher who was not involved in the interview.

## 3. Findings

A total of 18 cancer patients with a mean age of 54.9 years (SD = 7.3, range: 41–67) were interviewed. The largest proportion of participants were male (61.2%), with an educational level of high school (50.0%), married (66.7%), living with family members (94.4%), believed in Buddhism (38.9%), had a full-time job (38.9%), and had their spouse as the main caregiver (38.9%) (Table 1).

In terms of disease characteristics, the largest proportion suffered from head and neck cancer (38.9%), and the disease had metastasized to other sites in most cases (72.2%). A pain frequency of “always” (44.4%) was most common and an average level of worst pain of 8.6 (SD = 1.7, range: 5–10) was reported. Half the patients (50.0%) had been prescribed opioids on both ATC and PRN basis. Converting patients’ daily opioid prescriptions to morphine milligram equivalent yielded an average dose of 74.0 (SD = 39.0, range: 18.8–180.0). All patients faced opioid side effects. Twelve participants had a KPS (Karnofsky Performance Scale) score ≥ 60 (ranging from 0 to 100), with a higher score indicating better physical functioning in daily activities (Table 2). The psychometric properties of KPS have been confirmed [27].

This study employed inductive content analysis to understand participants’ experience of using non-pharmacological strategies to manage cancer pain. Five major themes emerged in this analysis: finding complementary therapies, utilizing various assistive skills, adapting to assistive skills, diverting attention, and seeking help. These themes reflected the study participants’ behavioral adaptations to manage their pain using non-pharmacological strategies. Each theme comprises several categories of behaviors. The participants faced situational barriers such as obstacles encountered in the workplace or situations affected by the climate. (See Table 3).

### 3.1. Theme One: Finding Complementary Therapies

When the participants dealt with cancer pain, in addition to taking painkillers, many sought complementary treatments. For example, cancer patients often sought traditional Chinese medicine (TCM) in conjunction with conventional treatments.

#### Seeking TCM Complementary Treatment

Many participants sought TCM treatments, such as acupuncture, tuina, and cupping, in hopes of reducing their pain. According to the majority of participants, TCM complementary treatments effectively reduced their pain, and the relief could last for as much as a day. For example, a participant recounted,

“*I go to a Chinese medicine clinic for tuina, hot compresses, and hot steaming for about 20–30 min at a time. The treatment is effective in relieving pain*”(Patient 2). Another participant stated,

“*During each acupuncture session, the TCM physician would insert 2 to 3 needles into the masticatory muscle of my right cheek for about 20 min each time. My mouth muscles become more relaxed and less painful*”(Patient 3). Another patient noted,

“*Sometimes when I am in pain, I would go to a Chinese medicine practitioner to get help for the pain. The doctor would perform bloodletting cupping therapy on me for about 30 min per session, at my own expense. It relieves my pain for at least one day, but the pain resurfaces the following day*”(Patient 14).

### 3.2. Theme Two: Utilizing Assistive Skills

While the participants took oral painkillers to control cancer pain, they also utilized a combination of additional assistive skills to deal with the pain. Some patients combined two assistive skills and used them simultaneously. The non-pharmacological assistive techniques employed by the participants included hot compresses, cold compresses, massage, proper body positioning, and a combination of various techniques.

#### 3.2.1. Applying Hot Compresses

Some participants used hot towels to compress the painful areas of their body, while others soaked their feet in hot water. Following a certain duration of heat application, some participants perceived that their pain was effectively relieved. The participants stated,

“*When I experience a headache, my initial response is to use a hot towel as a compress and apply it directly to the affected area. I keep the compress on each area for approximately a minute, beginning with the forehead, chin, and cheeks, and gradually moving downwards…*”(Patient 18);

“*When I take a bath, I frequently apply a hot towel as a compress to painful areas of the body, such as my back. To experience less pain, I need to keep the compress on for more than 10 min each time*”(Patient 2). One participant shared,

“*Sometimes, I use a hair dryer to blow hot air to my body after taking a shower for about half an hour. Afterward, my body feels very relaxed without any pain, and I also sleep better*”(Patient 1). One other participant noted,

“*… I usually soak my feet in hot water for about half an hour each time, and afterward, my entire body feels free from tension and pain*”(Patient 6).

#### 3.2.2. Applying Cold Compresses

The participants felt that cold compresses could produce a numbing effect, which can be particularly helpful for swelling and tingling pain. One participant said,

“*Patients like me who have tongue cancer often suffer from jaw pain. Cold compresses are ineffective for throbbing pain and numbness, but they occasionally can be very helpful for swelling pain with a burning sensation. It can sometimes provide relief for stinging pain, such as an ant bite or bee sting*”(Patient 3).

“*I used to put ice cubes in a plastic bag and applied it to my chin when I experienced pain. The effect was not bad; at least it alleviated my pain to some degree*”(Patient 4).

#### 3.2.3. Applying Massage

In the face of pain, some participants adopted massage strategies to relieve pain such as foot massage or body massage. For example,

“*At times, I can’t sleep well because of pain in the middle of the night, so I would ask my friend to give me a massage… applying pressure to the feet is quite effective, given that many acupuncture points are located on the feet*”(Patient 10). Patients feel more relaxed after massage. One participant stated,

“*Sometimes, because of the pain, I would get a foot massage, and the masseur would apply oil prior to the massage… I typically feel very relaxed afterward*”(Patient 15). Another participant said,

“*Whenever I am in pain, I visit a health club for massage therapy… my muscles are very relaxed during the session, which lasts about an hour. After receiving pressure applied to all my muscles, my entire body feels less painful*”(Patient 17).

#### 3.2.4. Using Body Positioning

The participants tried to utilize proper body positioning to relieve or prevent pain such as avoiding involvement of diseased areas. One participant explained,

“*I can only sleep sitting up at night, because cancer has metastasized to my neck. If I lie flat, my neck stretches, which causes pain. Hence, I can only sit upright and sleep sitting up*”(Patient 12). Another patient stated,

“*…my left shoulder blade hurts, so I always sleep on my right side. I have to adjust my sleeping position every night to ensure that I sleep on my right side. Otherwise, I would be in pain*”(Patient 7).

#### 3.2.5. Combining Multiple Assistive Skills

When a single non-pharmacological assistive skill failed to produce the desired results, the participants attempted to combine two assistive skills to relieve their pain. One participant stated,

“*…because of the pain, I applied ice to my cheeks every time… If the pain cannot be relieved, I quickly take painkillers. In addition to taking painkillers, I try to relieve the pain with ice cubes in my mouth…”*(Patient 9). One patient noted,

“*Sometimes, when it hurts too much, I sleep on my right side. At times, the left side still hurts, and the pain is unbearable, and my wife will take the initiative to give me a massage…*”(Patient 13). Another participant shared,

“*I first lie on my stomach in a fixed position, because this position is more comfortable. Before going to bed, I would ask my kids to give me a massage and apply a heat compress so that I can sleep better. Sleeping on the left side seems to press on my tumor, which hurts, so I must sleep on my right side*”(Patient 18).

### 3.3. Theme Three: Adapting Assistive Skills

Adaptation refers to flexibility in adjusting to different situations. The participants typically used familiar skills to manage pain, but when faced with situational changes, such as leaving the house or encountering unexpected obstacles, they might not be able to employ the usual skills. In these cases, they needed to be adaptable and apply alternative assistive skills in response to the situation and effectively manage the pain.

#### 3.3.1. Adapting Assistive Skills Outside the House

The participants adapted assistive skills to manage pain when faced with changes in context or situation, such as leaving the house or going to work. For example,

“*If I happen to be busy at work when the pain starts and I can’t go to the bathroom to spray on the pain medicine, I switch to patches. I put on six patches at once, two horizontally on the upper and lower back and two vertically on the front of the left and right thighs*”(Patient 1). One patient described his practices,

“*After undergoing radiation therapy, my neck was hot, scalded, and painful. It was very uncomfortable, especially when I went out. Ice water did not work well. Later I switched to ice towels and stored them in a plastic bag in the refrigerator. When I needed to leave the house, I would put them in a cooler. After radiation therapy, I used them as cold compresses to relieve the pain, which worked better*”(Patient 12). One patient stated,

“*…I initially tried to bring ice cubes and a towel with me to work to help relieve my pain… However, as I was already carrying tools, it was inconvenient to also bring ice cubes and a towel. Consequently, I stopped bringing ice cubes and switched to buying ice water instead. I drank ice water when I experienced pain and purchased more when I ran out…*”(Patient 9).

#### 3.3.2. Adapting Assistive Skills When Encountering Situational Barriers

When participants encountered challenging situations, they adopted flexible strategies to manage pain, for example due to climate or equipment challenges. One participant noted,

“*I take a walk to divert my attention from the pain, but on rainy days, it is inconvenient to go out for a walk… I would visit the supermarket to look at various products instead*”(Patient 4). Another participant said,

“*When I am in pain, sometimes I go for a jog in the park near my house to distract myself. If it rains, I opt for indoor swimming instead*”(Patient 11). One other participant stated,

“*I used to wear a corset under my clothes, and sometimes I would want to adjust the tightness of the corset when I didn’t feel the pain as much. Adjusting the tightness I have to do in a bathroom, which is quite inconvenient. I now wear the corset over my clothes, which makes it easy to adjust when necessary*”(Patient 10). Another patient stated,

“*I used to apply ice packs to my chin when it hurts, but this would leave my shirt wet in the chest area. So now I use special cold packs instead….*”(Patient 3).

### 3.4. Theme Four: Diverting Attention

The participants used various methods to divert their attention from pain. They usually engaged in activities of interest to distract themselves from pain, such as painting, video games, and mahjong, as described in these examples,

“*I can draw everything, oil painting, watercolor, sketching, and when I paint, I feel less pain because I have to concentrate on composing the picture and on thinking about how to match colors better, so I don’t have pain when I paint*”(Patient 6);

“*…I follow drama series to divert my attention from pain by watching TV. Sometimes I go out to play mahjong with friends. After all, when playing mahjong I have to be involved and pay attention to what tiles the other party would take and what tiles to wait for. I am not in pain when I play mahjong, so my wife encourages me to play…*”(Patient 3);

“*I play video games on my mobile phone. I play Candy Crush often and have passed over 4000 levels. When I concentrate on progressing through the levels, I have to think about navigating through obstacles in the game, so I don’t feel any pain*”(Patient 5).

### 3.5. Theme Five: Seeking Help

Some patients have limited physical capacity as a result of pain and may need assistance from family or friends to cope with pain and feel physically comfortable. The participants stated,

“*Whenever I have a headache, I ask my family members to give me a massage… they would massage my shoulders, neck, and hands. Once my muscles are relaxed, my headache seems to be less severe*”(Patient 18);

“*I use pain relief patches when I am in pain… however, it’s hard to apply the patch to my waist because I cannot reach the affected area… I finally asked my husband to help*”(Patient 2);

“*Sometimes when the pain is too intense, I can’t even get ice cubes from the refrigerator… I would ask my wife to help me apply an ice pack to my chin*”(Patient 3).

“*… I would ask my husband to rub my back gently from top to bottom… the amount of force has to be small. A ten-minute back rub each time would relieve some of my pain, and I would feel less painful*”(Patient 6).

## 4. Discussions

The findings of this study show that the interviewed patients combined non-pharmacological strategies with pharmacological treatment to alleviate their pain when faced with discomfort. These strategies included seeking complementary therapies, applying various assistive skills, and diverting their attention. When patients encountered situational obstacles while using non-pharmacological methods, they adapted to their assistive skills and sought help to relieve their pain. Although the use of analgesic drugs is the main method to manage moderate to severe cancer pain, this may not be completely effective. Additionally, some patients have concerns about the side effects of analgesic drugs or may only experience milder levels of pain [28]. In such cases, some patients choose TCM, such as acupuncture and cupping, to help alleviate pain. TCM reduces patients’ concerns related to addiction or tolerance issues that can arise from long-term drug use [29]. Acupuncture supplemented with drug therapy may yield faster pain relief, longer duration of relief, and better quality of life [30]. Acupuncture applied to the “si guan xue” point on the arm or to auricle points can effectively reduce pain [31]. A systematic literature review concluded that acupuncture effectively relieved pain among cancer patients in hospice palliative care wards [32], with no serious adverse side effects [30]. Other studies have suggested that cupping can be used alongside analgesic drugs to enhance pain relief in patients [33].

Although medication is the main strategy for managing moderate to severe cancer pain, we found that the interviewed patients used a variety of assistive skills to enhance pain relief, such as hot and cold compresses, massage, body positioning, and a combination of these. Other studies have also concluded that patients often integrate various non-pharmacological strategies such as hot compresses, posture changes, and even prayer, with pharmacological strategies to enhance pain relief [34]. Taking massage as an example, a study of RCT on massage measures [15] found that the pain intensities of patients in the experimental group were significantly lower than those of the control group in both the “pain at its worst” and “pain right now” categories. Previous studies have demonstrated that massage therapy can effectively relieve pain in patients [35]. Among three evaluated types of massage (foot reflexology, body massage, and aromatherapy massage), foot reflexology has been identified as the most effective for improving pain in patients [36].

The participants of the present study used hot compresses, hot foot baths, or cold compresses to help relieve pain. Past studies have suggested that the use of warm foot baths was able to significantly reduce the pain intensity of patients [17] and that there was support for the application of cold compresses [37]. In addition, the participants of the present study used body positioning to alleviate pain, a method also supported by previous research findings [37]. Appropriate body positioning and posture can improve blood circulation and alleviate muscle tension [38] and can be easily facilitated by family members to provide comfort and pain relief to terminally ill patients [39].

Diverting attention may temporarily block the transmission of pain stimuli [40]. Various activities such as reading, watching movies and TV [37], engaging in handicrafts [34], listening to music [41], and generally participating in favorite activities [42] can help to distract from pain. Studies have confirmed the effectiveness of music in enhancing pain relief [43]. The participants of the present study also used these methods frequently, and some participants diverted their attention from pain by interacting with others or with pets.

Moderate to severe pain can significantly interfere with the daily life of patients [44,45]. When daily activities are disrupted, the use of assistive skills may require the support of family and friends, particularly in cases where cancer patients are experiencing pain, fatigue, and physical exhaustion [46]. The assistance of relatives and friends plays an important role in the pain management of patients [37]. As evidenced in the present study, participants needed support from family or friends when utilizing non-pharmacological assistive skills to manage pain.

Patients may experience barriers to using relevant assistive skills due to changes in circumstances. We found that when participants were outside of their homes, such as in the workplace or in a climate-affected environment, their application of non-pharmacological strategies may be hindered or challenged. However, the participants were able to use strategies to adapt their assistive skills to cope with situational barriers. The findings of this study are consistent with previous studies highlighting the need to adapt assistive skills to manage pain and the observation that participants tried to employ various strategies to overcome situational barriers to the sustained use of such skills [34], particularly in workplace settings. Supportive workplace environments have been shown to facilitate the return to work of cancer patients after treatment [47,48], while unmanaged physical symptoms impede work productivity [49]. The results of the present study may provide a reference for the development of workplace policies that are friendly to patients experiencing pain.

Patients participating in the present study used the non-pharmacological strategies that were most convenient and accessible. No patients mentioned using psychosocial interventions for pain management, such as mindfulness-based strategies or hypnosis intervention, although these methods have also been reported as having partial effects of pain relief [18,50]. A possible explanation is that mindfulness-based strategies may require environmental preparation and additional skills and must initially be led by professionals. In addition, it may be that the non-pharmacological strategies, which were the primary focus of the semi-structured interview guides for the present study, may have caused patients to not think about psychosocial approaches.

Adequate drug prescriptions and patients’ compliance with these prescriptions in combination with non-pharmacological strategies may affect patients’ pain management outcomes. Pharmacological strategies to control cancer pain include opioids and adjuvants. The treatment of a transitory increase in pain includes regular opioid analgesics, together with additional opioids as needed. In the present study, we recorded patients’ opioid prescriptions but not adjuvants, and did not collect information about the patients’ actual opioid analgesics use. Even when regularly using opioid analgesics, patients still reported high levels of worst pain. These high reported pain levels may have been partly caused by the inclusion criteria of the study, which included a criterion of the patient suffering from a worst pain with a rating of at least 4 points on a 0–10 scale.

All participating patients faced opioid side effects, which might constitute one of the reasons that prompted patients to employ additional non-pharmacological strategies to manage pain. The clinical practice guidelines for cancer pain care developed by the National Comprehensive Cancer Network [12] recommend a combination of pharmacological and non-pharmacological strategies to manage cancer-related pain; as noted above, empirical evidence confirms this approach with respect to methods such as massage, hot or cold compresses, acupuncture, and cupping [15,17,30,32,33,37]. Particularly when patients are worried about medication, non-pharmacological strategies can increase the patient’s sense of autonomy over pain management and enhance pain relief. If the healthcare team can identify relevant non-pharmacological strategies for pain management and for overcoming challenges or obstacles that patients may face when using them, as well as teach and assist patients in adapting relevant skills to cope with obstacles, the behaviors of patients in using non-pharmacological complementary strategies for pain management may be enhanced.

## 5. Limitations

The sampling criteria for the present study included the stipulation that patients currently and regularly used prescribed opioid analgesics; however, we did not collect further data on adjuvants, and were thus unable to confirm whether the used drugs were prescribed appropriately. The present findings also cannot explain whether patients’ use of non-pharmacological strategies was able to reduce their drug use. Furthermore, the results of this study were obtained from a sample of 18 patients who experienced moderate pain and utilized non-pharmacological strategies for managing cancer pain. Participants in the study had different types of cancer, with the majority having head and neck cancer and distant metastases and were regularly using prescribed opioid analgesics. It is important to note that the relatively small sample size and the fact that participants were recruited from only one medical center in northern Taiwan may limit the generalizability of the findings of this study.

## 6. Conclusions

This study may provide valuable insights for healthcare teams in understanding how patients experiencing cancer pain utilize non-pharmacological strategies, the behavioral adaptations involved, and the situational barriers they may face in the process. Behavioral adaptations include finding non-pharmacological therapies, employing various assistive skills, adapting assistive skills, diverting attention, and seeking help in using assistive skills. Skills and behavioral adaptations related to non-pharmacologic strategies should be taught to patients with cancer pain, particularly those who have concerns about using painkillers or face situational barriers. These non-pharmacological strategies may complement patients’ pharmacological strategies to help manage cancer pain. This study identified situational barriers, such as being in the workplace or in a climate-affected environment, that may impede the use of non-pharmacological strategies, and these findings may be used to inform the development of relevant workplace policies that are friendly to cancer patients returning to work. Furthermore, the effectiveness of pain management should be regularly evaluated.

## Figures and Tables

**Table 1 healthcare-11-02911-t001:** Participant demographics (*n* = 18).

	Frequency	Percentage
Gender		
Male	11	61.2%
Female	7	38.8%
Age (years)	Mean = 54.9, SD = 7.3, range: 41–67	
Education		
College	4	22.2%
Junior college	2	11.1%
High school	9	50.0%
Middle school	2	11.1%
Elementary school	1	5.6%
Marital status		
Single	2	11.1%
Separated/divorced	4	22.2%
Married	12	66.7%
Living with friends or family		
Yes	17	94.4%
No	1	5.6%
Religion		
Buddhism	7	38.9%
Taoism	6	33.3%
Christian	1	5.6%
Other	1	5.6%
None	3	16.7%
Employment		
Full time	7	38.9%
Resigned	2	11.1%
Retired	7	38.9%
None	1	5.6%
Other	1	5.6%
Primary caregiver		
Spouse	7	38.9%
Parents	1	5.6%
Children	5	27.8%
Other	5	27.8%

**Table 2 healthcare-11-02911-t002:** Disease characteristics of participants (*n* = 18).

	Frequency	Percentage
Diagnosis		
Head and neck cancer	7	38.9%
Lung cancer	2	11.1%
Colorectal cancer	3	16.7%
Breast cancer	1	5.6%
Bladder cancer	4	22.2%
Other	1	5.6%
Distal metastasis		
Yes	13	72.2%
No	5	27.8%
Frequency of pain		
Sometimes	1	5.6%
Frequent	4	22.2%
Always	8	44.4%
Persistent	5	27.8%
Opioid prescription pattern		
ATC basis	18	100%
Both ATC and PRN basis	9	50%
Prescribed opioid type		
Oxycodone	12	66.7%
Morphine	8	44.4%
Fentanyl patch	7	38.9%
Tramadol	6	33.3%
MST	2	11.1%
MXL	1	5.6%
Daily prescribed opioid dose		
Morphine milligram equivalents	Mean = 74.0, SD = 39.0, range = 18.8–180.0	
Side effect of opioids		
Yes	18	100%
No	0	0.0%
Type of opioid side effect		
Drowsiness	8	44.4%
Dizziness	5	27.8%
Constipation	13	72.2%
Nausea	6	33.3%
Vomiting	6	33.3%
Dry mouth	4	22.2%
Sweating	4	22.2%
Pain level in the past week		
Level of worst pain	Mean = 8.6, SD = 1.7, range: 5–10	
Level of least pain	Mean = 2.6, SD = 1.0, range: 1–4	
Level of average pain	Mean = 5.1, SD = 1.4, range: 3–10	
Level of a transitory increase in pain	Mean = 7.7, SD = 2.1, range: 4–10	
KPS (0–100)		
≥60	12	66.7%
<60	6	33.3%

Note: KPS (Karnofsky Performance Scale); ATC (around the clock); PRN (as-needed basis).

**Table 3 healthcare-11-02911-t003:** Behaviors and related contexts of non-pharmacological strategies for cancer pain management.

(Theme)	(Category)
Finding complementary therapy	Seeking traditional Chinese medicine treatment
Utilizing assistive skills	Applying hot compress
	Applying cold compress
	Applying massage
	Using body positioning
	Combining multiple assistive skills
Adapting to assistive skills	Adapting to assistive skills outside of the house
	Adapting to assistive skills for situational barriers
Diverting attention	Adopting various methods to divert attention
Seeking help	Seek help for using assistive skills

## Data Availability

Data presented in this study are available from the first author upon reasonable request.

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
