# Peer review of "The Behavioral Adaptations and Barriers of Patients Employing Non-Pharmacological Strategies for Cancer Pain Management—A Qualitative Study"

_healthcare, 2023, doi:10.3390/healthcare11222911_

Round 1
Reviewer 1 Report (Previous Reviewer 1)
Comments and Suggestions for Authors
Thank you for including me as a reviewer for this next round. This article investigated behavioural pain management strategies alongside pharmacological approaches. Again I find this article is high quality with minor edits suggested.
Follow-up from previous suggestions that still require attention:
1 - In regards to literature on psychosocial interventions, mindfulness is only one. Here is a meta-analysis on such interventions on pain management for patients with cancer, please add it when discussing psychosocial interventions:
Gorin SS et al. (2012). Meta-analysis of psychosocial interventions to reduce pain in patients with cancer. Journal of Clinical Oncology 30(5): 539-547. doi: 10.1200/JCO.2011.37.0437.
I also wonder if your interview guided participants to think about behavioural interventions, which may have not made them think about psychotherapies. Perhaps mention that this could be an explanation for the lack of findings of such strategies.
Also, good hypothesis on the barrier of mindfulness intervention (page 11, lines 403-405).
2- When describing the sample in Methods, please include the word inpatients: "This study took place on an inpatient oncology ward..."
3- Regarding the strategy of using a corset- I didn't mean for you to remove this. I was simply commenting that I didn't know about this strategy. I wasn't suggesting any change.
New comments:
4- In the abstract, line 26, there is a typo "no-pharmacological" rather than "non-".
5- The final line in the abstract "Thus, health and social [...] assess effectiveness of the strategies" does not fit with the conclusions in your paper. While they aren't in your paper, I do think these are important to be discussing. Your findings highlight the importance of interdisicplinary teams, including medical, psychosocial (psychology, social work, spiritual care), and rehabilitation professionals.
6- Page 6, line 181- "non-drug" could be changed to "non-pharmacological" to keep consistency throughout the paper.
7- Page 10, line 388- "We found" as opposed to "we showed."
Overall, I think this manuscript is very good and almost ready for publication. It will make an excellent addition to the literature.
Comments on the Quality of English LanguageMostly fine, some minor wording that editing will likely help with.
Author Response
Comments and Suggestions for Authors
Thank you for including me as a reviewer for this next round. This article investigated behavioural pain management strategies alongside pharmacological approaches. Again I find this article is high quality with minor edits suggested.
Follow-up from previous suggestions that still require attention:
1 - In regards to literature on psychosocial interventions, mindfulness is only one. Here is a meta-analysis on such interventions on pain management for patients with cancer, please add it when discussing psychosocial interventions:
Gorin SS et al. (2012). Meta-analysis of psychosocial interventions to reduce pain in patients with cancer. Journal of Clinical Oncology 30(5): 539-547. doi: 10.1200/JCO.2011.37.0437.
I also wonder if your interview guided participants to think about behavioural interventions, which may have not made them think about psychotherapies. Perhaps mention that this could be an explanation for the lack of findings of such strategies.
Also, good hypothesis on the barrier of mindfulness intervention (page 11, lines 403-405).
Response: We add Gorin SS et al. (2012) in discussing psychosocial interventions. In addition, we add more explanations for the lack of findings of psychosocial strategies.
2- When describing the sample in Methods, please include the word inpatients: "This study took place on an inpatient oncology ward..."
Response: Yes, we add the word “inpatient”.
3- Regarding the strategy of using a corset- I didn't mean for you to remove this. I was simply commenting that I didn't know about this strategy. I wasn't suggesting any change.
Response: We put in this patient statement.
New comments:
4- In the abstract, line 26, there is a typo "no-pharmacological" rather than "non-".
Response: We fixed this typo.
5- The final line in the abstract "Thus, health and social [...] assess effectiveness of the strategies" does not fit with the conclusions in your paper. While they aren't in your paper, I do think these are important to be discussing. Your findings highlight the importance of interdisicplinary teams, including medical, psychosocial (psychology, social work, spiritual care), and rehabilitation professionals.
Response: We change “health and social professionals” to “health professionals”.
6- Page 6, line 181- "non-drug" could be changed to "non-pharmacological" to keep consistency throughout the paper.
Response: We change "non-drug" to "non-pharmacological".
7- Page 10, line 388- "We found" as opposed to "we showed."
Response: We change "we showed" to "We found".
Overall, I think this manuscript is very good and almost ready for publication. It will make an excellent addition to the literature.
Response: Thank you very much for helping us to improve our manuscript.

Reviewer 2 Report (Previous Reviewer 2)
Comments and Suggestions for Authors
The paper is much improved in the present version, and is now suitable for publication, just give a look to English writing and punctuation.
Comments on the Quality of English LanguageJust check all English and punctuation, but in general also the quality of English is improved.
Author Response
Comments and Suggestions for Authors
The paper is much improved in the present version, and is now suitable for publication, just give a look to English writing and punctuation.
Response: Thank you very much for helping us to improve our manuscript.

Reviewer 3 Report (Previous Reviewer 3)
Comments and Suggestions for Authors
The work has been partially rewritten taking into account the referees' observations. Therefore in the current version it can be published in the journal in question. There are some limitations in the work that are well explained in the section 5 "Limitations".
Author Response
Comments and Suggestions for Authors
The work has been partially rewritten taking into account the referees' observations. Therefore in the current version it can be published in the journal in question. There are some limitations in the work that are well explained in the section 5 "Limitations".
Response: Thank you very much for helping us to improve our manuscript.

This manuscript is a resubmission of an earlier submission. The following is a list of the peer review reports and author responses from that submission.
Round 1
Reviewer 1 Report
Comments and Suggestions for Authors
Thank you for inviting me to review this paper. This is a qualitative analysis assessing barriers in using nonpharmacological strategies for pain reduction in patients with advanced cancer. This study yielded important and interesting results. I have some minor suggestions, and while they are minor, I believe they are important.
Introduction:
In the literature of cancer and palliative care, pain is known as a function of multiple domains of suffering, such as physical, psychological, social, spiritual, and others. Yet, in treating pain, interventions tend to be pharmacological as the authors have pointed out. However, the authors have missed including the literature on the mulitple sources of pain as an explanation for the limited effects of pharmacological interventions. Please add some literature on the multiple dimension of pain and the importance of non-pharmacological interventions for this reason (i.e. not all pain is due to physical tissue damage).
On page 1 line 41, "still reported persistent pain…." verses "persisted pain."
There is a noticeable lack of literature review on psychosocial interventions for pain. Perhaps this is a cultural difference, I aologize if so. In the academic literature, psychosocial interventions for pain are well documented, although interestingly poorly implemented in Western medical care. I am unaware if psychosocial interventions by psychologists, social workers, psychiatrists, or spiritual care practitioners would be a regular part of cancer care in the Eastern medical system. I would suggest including at least a few sentences on psychosocial interventions for pain in the introduction, such as mindfulness-based cancer recovery, or other psychological pain management strategies that are present in the academic literature. It is notable that such interventions were not mentioned by participants, assuming that such interventions might be offered as part of cancer care in the East. If so, the lack of mention is notable in the discussion, as well.
On page 2 line 68, I believe you meant "physical activity," as opposed to "physical inactivity."
Interesting research question.
Methods
Are these inpatients?
Please describe "breakthrough pain." Not all readers will understand this term yet.
On page 2 line 89. I believe you can end the last sentence after "stopped when no new categories could be derived." This is a fairly standard approach in qualitative research and this wording is sufficient.
Looking at the questions asked, as described in the second graph on page 3, I wonder if the patients understood these questions, the wording seems technical. Were explanations needed by the interviewers? Please include the full interview script, either in the text or as an appendix.
On page 3 line 109, "after the interview, recordings are transcribed verbatim and analysed within 48 hours." I have removed the words "the interview" after the first comma and "it" after the word "analyzed."
Were responses translated to English for this manuscript? How was the integrity of the translation assured? Although the English in this article is quite good, there are some particular words that did not translate properly. I've mention them here, although a second review for minor errors would be helpful.
Results
On page 4 in the employment section, I would suggest combining the None and Other categories, since they are only one person each.
In the paragraph between table 1 and table 2, I would suggest including percentages rather than number of participants.
In table 2, there is extremely high levels of pain frequency and breakthrough pain. This might be important to note in the discussion, because even with some of the strategies, pain overall and breakthrough pain remain high.
The word climate is used throughout to describe the workplace environment. But "workplace environment" or "workplace atmosphere" might be a better tersm to use. Although climate is not technically wrong, it is not quite right in this context. When using this for the first time, perhaps provide some examples to anchor the reader.
On page 5, line 170 the word fumigation is used. This is not a proper translation in English, fumigation refers to the extermination of rodents and insects in a building or home.
On page 8 lines 278 onwards, there is discussion about using the corset under clothes. This is a pain management technique? Interesting. I didn't know.
Discussion
Pain remained high according to your results. Is there some discussion to add on what patients said works better or worse between the methods discussed?
On page 9 lines 370 onwards, the authors discuss improving the workplace environment to enhance strategy utilization. Perhaps the authors can say a few words on this and make suggestions of next steps, here. It is to the workplace's advantage to improve access for people with disabilities. Ultimately, it saves them money in reduced absenteeism and improved productivity. Perhaps this can be included as a future direction.
Overall, I thought this was a very interesting and important article thank you for including me as a reviewer.
Comments on the Quality of English LanguageSome minor changes noted
Author Response
Response to Reviewer 1 Comments
Comments and Suggestions for Authors
Thank you for inviting me to review this paper. This is a qualitative analysis assessing barriers in using nonpharmacological strategies for pain reduction in patients with advanced cancer. This study yielded important and interesting results. I have some minor suggestions, and while they are minor, I believe they are important.
Introduction:
In the literature of cancer and palliative care, pain is known as a function of multiple domains of suffering, such as physical, psychological, social, spiritual, and others. Yet, in treating pain, interventions tend to be pharmacological as the authors have pointed out. However, the authors have missed including the literature on the mulitple sources of pain as an explanation for the limited effects of pharmacological interventions. Please add some literature on the multiple dimension of pain and the importance of non-pharmacological interventions for this reason (i.e. not all pain is due to physical tissue damage).
Response: We add “Pain is a multiple domain of suffering including physical, psychological, social, cognitive, and spiritual dimension” to connect pharmacological and non-pharmacological interventions
On page 1 line 41, "still reported persistent pain…." verses "persisted pain."
Response: We change "persisted pain" to "persistent pain"
There is a noticeable lack of literature review on psychosocial interventions for pain. Perhaps this is a cultural difference, I aologize if so. In the academic literature, psychosocial interventions for pain are well documented, although interestingly poorly implemented in Western medical care. I am unaware if psychosocial interventions by psychologists, social workers, psychiatrists, or spiritual care practitioners would be a regular part of cancer care in the Eastern medical system. I would suggest including at least a few sentences on psychosocial interventions for pain in the introduction, such as mindfulness-based cancer recovery, or other psychological pain management strategies that are present in the academic literature. It is notable that such interventions were not mentioned by participants, assuming that such interventions might be offered as part of cancer care in the East. If so, the lack of mention is notable in the discussion, as well.
Response: We add some sentences on psychosocial interventions for pain in the introduction (see p.2). We also discuss this finding is lacking of psychosocial interventions in the discussion section.
On page 2 line 68, I believe you meant "physical activity," as opposed to "physical inactivity."
Response: We change "physical inactivity" to "physical activity".
Interesting research question.
Methods
Are these inpatients?
Response: Yes, the study took place at an oncology ward in a medical center in northern Taiwan (see 2.2. Sample and setting). These patients are all newly hospitalized. During the interviews, the patients were asked to provide their experiences at home.
Please describe "breakthrough pain." Not all readers will understand this term yet.
Response: We use parentheses to add colloquial descriptions of breakthrough pain.
On page 2 line 89. I believe you can end the last sentence after "stopped when no new categories could be derived." This is a fairly standard approach in qualitative research and this wording is sufficient.
Response: We end the last sentence after "stopped when no new categories could be derived."
Looking at the questions asked, as described in the second graph on page 3, I wonder if the patients understood these questions, the wording seems technical. Were explanations needed by the interviewers? Please include the full interview script, either in the text or as an appendix.
Response: We've added additional interview scripts.
On page 3 line 109, "after the interview, recordings are transcribed verbatim and analysed within 48 hours." I have removed the words "the interview" after the first comma and "it" after the word "analyzed."
Response: We remove "the interview" and "it" in the sentence.
Were responses translated to English for this manuscript? How was the integrity of the translation assured? Although the English in this article is quite good, there are some particular words that did not translate properly. I've mention them here, although a second review for minor errors would be helpful.
Response: We correct some particular words that did not translate properly based on reviewers’ suggestions.
Results
On page 4 in the employment section, I would suggest combining the None and Other categories, since they are only one person each.
Response: We would like to keep the None and Other categories. Maintaining original data may aid in the interpretation of information.
In the paragraph between table 1 and table 2, I would suggest including percentages rather than number of participants.
Response: We change the number of participants to percentage of participants.
In table 2, there is extremely high levels of pain frequency and breakthrough pain. This might be important to note in the discussion, because even with some of the strategies, pain overall and breakthrough pain remain high.
Response: We discuss the high levels of pain frequency and breakthrough pain in discussion section.
The word climate is used throughout to describe the workplace environment. But "workplace environment" or "workplace atmosphere" might be a better terms to use. Although climate is not technically wrong, it is not quite right in this context. When using this for the first time, perhaps provide some examples to anchor the reader.
Response: The word "climate" in the text represents weather. In addition to the obstacles of the working environment, there are also obstacles of the weather. “…in the workplace or in a climate-affected environment…” is “…in the workplace or in a weather-affected environment…”.
On page 5, line 170 the word fumigation is used. This is not a proper translation in English, fumigation refers to the extermination of rodents and insects in a building or home.
Response: We change the word “fumigation” to “hot steaming”.
On page 8 lines 278 onwards, there is discussion about using the corset under clothes. This is a pain management technique? Interesting. I didn't know.
Response: We delete this example. Actually, we truthfully present patients’ descriptions of their pain management.
Discussion
Pain remained high according to your results. Is there some discussion to add on what patients said works better or worse between the methods discussed?
Response: One of the inclusion criteria in this study is that the patient has breakthrough pain. Breakthrough pain is a transitory flare of pain in the setting of chronic pain managed with opioid drugs. We have added some reminders in the discussion to show that severe pain still relies on drugs as a mainstay and non-drugs as a supplement. It is also added that the current study did not collect the patients' opioid prescriptions, and did not understand the patients' actual opioid analgesics use. Adequate drug prescriptions and patients' compliance with the drugs combines non-pharmacological strategies may impact on patients' pain management outcomes.
On page 9 lines 370 onwards, the authors discuss improving the workplace environment to enhance strategy utilization. Perhaps the authors can say a few words on this and make suggestions of next steps, here. It is to the workplace's advantage to improve access for people with disabilities. Ultimately, it saves them money in reduced absenteeism and improved productivity. Perhaps this can be included as a future direction.
Response: we add “Unmanaged physical symptom hinders work productivity [42]”.
Overall, I thought this was a very interesting and important article thank you for including me as a reviewer.

Reviewer 2 Report
Comments and Suggestions for Authors
This paper wants to explore the barriers for non pharmacological approach to pain in cancer pain.
However, there is no definition of pain (intensity, quality, duration, pathophysiology). Moreover, breakthrough pain is not consistent with current definition. No mention is done on about the pharmacological treatment for cancer pain (drugs, doses, and potential adverse effect.)
No mention was made about how many patients used every complementary strategy or why there prefer one or another.
The study enrolled only 18 patients, and it just reported the experience of the small group of patients. Therefore, it seems more a case series than qualitative research, since there is no hypothesis on how complementary strategies work and why some patients adopt them with a pharmacological approach.
In the discussion, the finding of the present study are just reported, with similar data available in literature, but no critical insight, nor hypothesis on treatments, their effectiveness, their potential limits and bias.
Also potential barriers are just provided but there is no discussion on their causes or on the possibility to remove them.
The study simply describes the experience of this limited group of patients. The data are highly suggestive about the utility of non pharmacological strategy to control cancer pain, I agree, but, once again, in my opinion the present paper is an interesting case series .
Comments on the Quality of English Language
The paper has no method, is just a case series. Moreover, pain is not assessed, measured, evaluated.
The proposed strategies are well-known and the paper does not add no further insights about them, since there is no mention about how many patients used every complementary strategy or why there prefer one or another.
Author Response
Response to Reviewer 2 Comments
Comments and Suggestions for Authors
This paper wants to explore the barriers for non pharmacological approach to pain in cancer pain.
However, there is no definition of pain (intensity, quality, duration, pathophysiology). Moreover, breakthrough pain is not consistent with current definition. No mention is done on about the pharmacological treatment for cancer pain (drugs, doses, and potential adverse effect.)
Response: We use parentheses to add colloquial descriptions of breakthrough pain “(a transitory increase in pain even though taking pain medicine regularly for cancer pain)”. We did not collect the type or dose of opioid prescribed of participants. Our sampling conditions considered patients with regular opioid use, and the main focus of this study was on non-pharmacological strategies. Our inclusion criteria are cancer patients who (1) had experienced pain for more than 3 months, (2) suffered from breakthrough pain (a transitory increase in pain even though taking pain medicine regularly for cancer pain) with a rating of 4 points (inclusive) or more on a 0–10 scale in the past week, (3) currently and routinely use prescribed opioid analgesics…
No mention was made about how many patients used every complementary strategy or why there prefer one or another.
Response: Our main focus is to explore the behavioral adaptations and situational barriers that cancer patients encounter when utilizing non-pharmacological strategies to manage pain. In the content analysis process, we did not count the frequency of patients used every complementary strategy.
The study enrolled only 18 patients, and it just reported the experience of the small group of patients. Therefore, it seems more a case series than qualitative research, since there is no hypothesis on how complementary strategies work and why some patients adopt them with a pharmacological approach.
Response: We based on the literature to add “Pain is a multiple domain of suffering including physical, psychological, social, cognitive, and spiritual dimension…” to connect pharmacological and non-pharmacological interventions in the introduction section. For qualitative research, we have tried our best to explain the research process in the research method section, including revising our manuscript according to the reviewers’ suggestions. The findings of the current study can provide reference for clinical practice of cancer pain management.
In the discussion, the finding of the present study are just reported, with similar data available in literature, but no critical insight, nor hypothesis on treatments, their effectiveness, their potential limits and bias.
Response: We add new content to the discussion section based on reviewers' suggestions to strengthen this part.
Also potential barriers are just provided but there is no discussion on their causes or on the possibility to remove them.
Response: We have explored patients' experiences regarding the causes of obstacles and how they deal with the obstacles.
The study simply describes the experience of this limited group of patients. The data are highly suggestive about the utility of non pharmacological strategy to control cancer pain, I agree, but, once again, in my opinion the present paper is an interesting case series.
Response: For qualitative research, we have tried our best to explain the research process in the research method section, including revising the discussion section according to the reviewers’ suggestions.
Comments on the Quality of English Language
The paper has no method, is just a case series. Moreover, pain is not assessed, measured, evaluated.
The proposed strategies are well-known and the paper does not add no further insights about them, since there is no mention about how many patients used every complementary strategy or why there prefer one or another.
Response: We response these questions above.
Reviewer 3 Report
Comments and Suggestions for Authors
This is qualitative research that uses inductive content analysis to explore patients' experiences.
There are some limitations that must be overcome to make the publication acceptable for publication.
The English language is grammatically correct, but the sentences are convoluted: a revision by a native translator would be appropriate.
Line 88 = this sentence needs to be made more understandable
Lines 117 and following = the parameters defined are arbitrary, even if reported in the literature, and the definitions are often self-referential. This point should be explained better, also within its limits of transferability of the results which is very dependent on the choices made by the researchers.
Line 161 and following: The “themes” are defined by sentences reported by patients. The anecdotal value of this collection methodology must be underlined.
Lines 386 and following: Limitations: the small sample size is rightly underlined. Furthermore, another aspect emerges that should be included to better understand the effectiveness of the interventions adopted: has the use of complementary strategies significantly changed the use of opioids? This could be a very interesting indicator.
Comments on the Quality of English LanguageThe English language is grammatically correct, but the sentences are convoluted: a revision by a native translator would be appropriate.
Author Response
Response to Reviewer 3 Comments
Comments and Suggestions for Authors
This is qualitative research that uses inductive content analysis to explore patients' experiences.
There are some limitations that must be overcome to make the publication acceptable for publication.
The English language is grammatically correct, but the sentences are convoluted: a revision by a native translator would be appropriate.
Response: We correct some sentences based on the suggestions of reviewers.
Line 88 = this sentence needs to be made more understandable.
Response: We revise this sentence to “A sufficient sample size of eighteen patients was determined based on the data saturation of findings”.
Lines 117 and following = the parameters defined are arbitrary, even if reported in the literature, and the definitions are often self-referential. This point should be explained better, also within its limits of transferability of the results which is very dependent on the choices made by the researchers.
Response: We add more our approaches to the trustworthiness of this research.
Line 161 and following: The “themes” are defined by sentences reported by patients. The anecdotal value of this collection methodology must be underlined.
Response: Yes, some codes can become themes themselves if they are interesting or classic words.
Lines 386 and following: Limitations: the small sample size is rightly underlined. Furthermore, another aspect emerges that should be included to better understand the effectiveness of the interventions adopted: has the use of complementary strategies significantly changed the use of opioids? This could be a very interesting indicator.
Response: We add “The sampling criteria for the current study were patients who were currently and regularly using prescribed opioid analgesics, so these non-pharmacological strategies were not used in addition to pharmacological strategies. The current findings also cannot explain whether patients' use of non-pharmacological strategies can reduce their drug use.” in research limitation section. For patients with moderate to severe pain (level 4-10 on a 0–10 scale), the main use of analgesic drugs is still supplemented by complementary strategies. However, in patients with mild pain (under level 3), complementary strategies alone may be acceptable.
Comments on the Quality of English Language
The English language is grammatically correct, but the sentences are convoluted: a revision by a native translator would be appropriate.
Response: We correct some sentences based on the suggestions of reviewers.
Reviewer 4 Report
Comments and Suggestions for Authors
Thank you very much for allowing me to review this manuscript that addresses a topic of interest to society. I would like to make a series of considerations regarding the publication of the manuscript:
Keywords: a keyword could be included that defines the type of study methodology used.
Introduction:
* I think it should be discussed in the introduction section of the WHO analgesic ladder, which includes what they mention: analgesics + non-pharmacological measures. They reference the WHO, but the ladder should be specifically cited, since it was created specifically for the treatment and approach to cancer pain.
* It is not clear what the objective is at the end of the introduction. Please indicate clearly.
Methods:
*One of the inclusion criteria is the level of pain that the person suffers. Please indicate which instrument or scale was used to measure the intensity of pain in each person.
*Was the type or dose of opioid prescribed taken into account?
* Was any computer program used for transcription or analysis of the speech?
* Include statistical information on the psychometric characteristics of the KPS scale? What does it measure? Etc…
Discussion:
* They should be careful when making categorical statements. In the first sentence they say the following: “The findings of this study showed that cancer patients used non-pharmacological 313 strategies to alleviate their pain when faced with discomfort.” The sample size used in this study is small and the qualitative methodology does not allow making such categorical statements or extrapolating the results. These observations are limited only to the people who participated in the study. It should be taken into account and reviewed throughout the discussion.
On the other hand, throughout it it seems that more effectiveness is being assigned to pseudotherapies than to conventional pharmacological treatments. These therapies have not shown enough scientific evidence to make those claims. I recommend that you review the entire discussion and include these aspects in the limitations.
Conclusions:
* The same goes for the conclusions: “This study may provide valuable insights for healthcare teams in understanding how patients experiencing cancer pain utilize non-pharmacological strategies, the behavioral adaptations involved, and the situational barriers they may face in the process.” They must review the statements considering that they can only be limited to the participating people.
Author Response
Response to Reviewer 4 Comments
Comments and Suggestions for Authors
Thank you very much for allowing me to review this manuscript that addresses a topic of interest to society. I would like to make a series of considerations regarding the publication of the manuscript:
Keywords: a keyword could be included that defines the type of study methodology used.
Response: We add “qualitative research” to be a keyword.
Introduction:
* I think it should be discussed in the introduction section of the WHO analgesic ladder, which includes what they mention: analgesics + non-pharmacological measures. They reference the WHO, but the ladder should be specifically cited, since it was created specifically for the treatment and approach to cancer pain.
Response: We add the WHO analgesic ladder for the treatment and approach to cancer pain in the introduction section.
* It is not clear what the objective is at the end of the introduction. Please indicate clearly.
Response: We add the research objective at the end of the introduction.
Methods:
*One of the inclusion criteria is the level of pain that the person suffers. Please indicate which instrument or scale was used to measure the intensity of pain in each person.
Response: We shown “on a 0–10 scale” in the (2) of the inclusion criteria.
*Was the type or dose of opioid prescribed taken into account?
Response: Our main focus is to explore the behavioral adaptations and situational barriers that cancer patients encounter when utilizing non-pharmacological strategies to manage pain. We did not collect the type or dose of opioid prescribed of participants. Our inclusion criteria (3) is currently and routinely use prescribed opioid analgesics. The non-pharmacological strategies of the participants in this study were used to supplement their pharmacological strategies.
* Was any computer program used for transcription or analysis of the speech?
Response: We did not use any computer program for transcription or analysis of the speech for this qualitative study.
* Include statistical information on the psychometric characteristics of the KPS scale? What does it measure? Etc…
Response: We add psychometric properties of this scale in the paragraph (see p.5).
Discussion:
* They should be careful when making categorical statements. In the first sentence they say the following: “The findings of this study showed that cancer patients used non-pharmacological strategies to alleviate their pain when faced with discomfort.” The sample size used in this study is small and the qualitative methodology does not allow making such categorical statements or extrapolating the results. These observations are limited only to the people who participated in the study. It should be taken into account and reviewed throughout the discussion.
Response: We revise this sentence to “The findings of this study showed that patients in the current study combined non-pharmacological strategies to …” and taken into account and reviewed throughout the discussion.
On the other hand, throughout it seems that more effectiveness is being assigned to pseudotherapies than to conventional pharmacological treatments. These therapies have not shown enough scientific evidence to make those claims. I recommend that you review the entire discussion and include these aspects in the limitations.
Response: We add “The sampling criteria for the current study were patients who were currently and regularly using prescribed opioid analgesics, so these non-pharmacological strategies were not used in addition to pharmacological strategies.” in the limitation section. We also emphasize this sampling criteria in the discussion section. Most clinical studies have controlled drug strategies as interfering variables to support the efficacy of non-drug strategies. Therefore, non-drug strategies are often used as auxiliary strategies in clinical applications.
Conclusions:
* The same goes for the conclusions: “This study may provide valuable insights for healthcare teams in understanding how patients experiencing cancer pain utilize non-pharmacological strategies, the behavioral adaptations involved, and the situational barriers they may face in the process.” They must review the statements considering that they can only be limited to the participating people.
Response: We add “from findings of the present study”. We also add “These nonpharmacological strategies may complement patients' pharmacological strategies to help manage cancer pain” in Conclusions section.

Round 2
Reviewer 2 Report
Comments and Suggestions for Authors
The paper is very similar to first version. Breaktrough cancer pain is a severe and short lasting attack of pain (NRS>7) in a baseline pain that is adequately controllled.
In your case series, 44% of patient reported pain as always present, and 27% as persistent, so baseline pain is not adequately controlled.
Therefore, are we speaking about 18 patients with BTcP or with baseline pain no controlled at all?
Moreover, how you can state how is pain intensity, without measuring it?
The problem of not analyzing or reporting drug therapy is a big limit in my opinion, since we have no idea of how severe is this pain, if the patients could increase doses, if they do adjuntive therapies (corticosteroids for example, anticonvulsants).
How is sample size calculated? The text is not clear at all.
The rigor of a study relies on the methods section, the fully explanation of inclusion and exclusion criteria, a rigorous statistical method (not declared here) and a wide and consistent data collection.
Moreover, why do you state in limits that pharmacological and non pharmacological strategies were not used in addition? In the methods you state that all patients are regularly on oppiod therapy!
The fact that you did not collect data on therapy is a limit in my opinion and should be stressed.
Comments on the Quality of English LanguageThe english need some more improvents , for example line 77 is not clear.
Reviewer 4 Report
Comments and Suggestions for Authors
Thank you very much for allowing me to review the manuscript again. The work has improved significantly and the authors have addressed all the raised issues favorably, so there are no further questions.
